# Novel Application of Mineral By-Products Obtained from the Combustion of Bituminous Coal–Fly Ash in Chemical Engineering

**Eleonora Sočo [1],\* , Dorota Papciak [2] and Magdalena M. Michel [3]**

[1] Department of Inorganic and Analytical Chemistry, Faculty of Chemistry, Rzeszów University of Technology, 6 Powstańców Warszawy Ave., 35-959 Rzeszów, Poland

[2] Department of Water Purification and Protection, Faculty of Civil, Environmental Engineering and Architecture, Rzeszów University of Technology, 6 Powstańców Warszawy Ave., 35-959 Rzeszów, Poland; dpapciak@prz.edu.pl

[3] Institute of Environmental Engineering, Warsaw University of Life Sciences-SGGW, Nowoursynowska 166, 02-787 Warsaw, Poland; magdalena_michel@sggw.pl

\* Correspondence: eleonora@prz.edu.pl; Tel.: +48-17-865-1508

**Abstract:** The aim of this work was the chemical modification of mineral by-products obtained from the combustion of bituminous coal (FA) treated with hydrogen peroxide (30%), used as an adsorbent for the removal of Cr(III) and Cd(II) ions and crystal violet (CV) from a mixture of heavy metal and organic dye in a solution containing either Cr(III)–CV or Cd(II)–CV. Fourier transform infrared (FT-IR), thermogravimetric analysis (TG), scanning electron microscopy–energy dispersive spectroscopy (SEM-EDS), and X-ray diffraction (XRD) analyses suggested that the mechanism of Cr(III)–CV or Cd(II)–CV sorption onto FA–$H_2O_2$ includes ion-exchange and surface adsorption processes. The effect of pH on the adsorption equilibrium was studied. The maximum adsorption was found for pH values of 9. The values of the reduced chi-square test ($\chi^2$/degree of freedom (DoF)) and the determination coefficient $R^2$ obtained for the sorbate of the considered isotherms were compared. Studies of equilibrium in a bi-component system by means of the extended Langmuir (EL), extended Langmuir–Freundlich (ELF), and Jain–Snoeyink (JS) models were conducted. The estimation of parameters of sorption isotherms in a bi-component system, either Cr(III)–CV or Cd(II)–CV, showed that the best-fitting calculated values of experimental data for both sorbates were obtained with the JS model (Cr(III) or CV) and the EL model (Cd(II) or CV). The maximum monolayer adsorption capacities of FA–$H_2O_2$ were found to be 775, 570 and 433 mg·g$^{-1}$ for Cr, Cd and CV, respectively. Purification water containing direct Cr(III) or Cd(II) ions and CV was made with 90%, 98% and 80% efficiency, respectively, after 1.5 h. It was found that the chemical enhancement of FA from coal combustion by $H_2O_2$ treatment yields an effective and economically feasible material in chemical engineering for the treatment of effluents containing basic dyes and Cr(III) and Cd(II) ions.

**Keywords:** mineral by-products; chemical engineering; sorption; bi-component system; crystal violet; heavy metals

## 1. Introduction

Growing amounts of by-products such as coal fly ash are generated every year from the combustion of bituminous coal in electric power plants [1]. A large portion of coal fly ash is disposed of in landfills [2,3]. Because of the low leaching potential of toxic elements, coal ash is widely used as a recycling material (250 million tons per year) [2–5].

Fly ash presents differentiated physicochemical and mineralogical characteristics, which are relevant for its potential application. The abundance of amorphous aluminosilicate glass is what makes fly ash an important source sorbent material [6–9]. Coal fly ash has potential as a source in the production of synthetic zeolite—the effective sorbent for nickel removal from water [10]. In recent years, there has been growing interest in the search for cheaper sources, such as low-cost adsorbent materials, for the immobilization of dyes and heavy metals from mixtures [11–17]. For this reason, there are studies on the utilization of this solid waste [18–22] for the removal of heavy metals [23–25] and dyes [26–28] from polluted waters.

The use of synthetic dyes is increasingly common in textile industries, dye manufacturing industries, paper and pulp mills, tanneries, electroplating factories, food companies, and others [29]. Therefore, it is necessary to study the adsorption phenomena of different classes of dyes. Crystal violet (CV) is a basic dye which belongs to the group of tri-phenylmethanes. It is a typical cationic dye that has been widely used as a colorant in textile products. Incrustation is the most visible effect of wastewater discharge from the textile industry [28,30]. The distribution of compounds containing Cr(III) and Cr(VI) depends on the redox potential. According to the World Health Organization (WHO), Cr(VI) and Cr(III) have different physicochemical properties: Cr(III) compounds are converted to mutagens only in a strong oxidizing environment, while Cr(VI) has carcinogenic and mutagenic properties [31]. It is most often accumulated in the lungs and kidneys. Cadmium, which is a human carcinogen, exerts toxic effects on the respiratory and the skeletal systems [32,33].

Several dyes and other effluents in chemical plants contain Cr(VI), which is much more hazardous than Cr(III). The most commonly used approach for the detoxification of hazardous industrial effluents and wastewaters containing Cr(VI) is its reduction to the much less toxic and immobile form of Cr(III) [34] and the identification of the main mechanisms involved in the removal of Cr(VI) [35].

Despite many studies on the immobilization of heavy metals, it is still important to improve our understanding of the relationship between the chemical composition of fly ash and its sorption properties [22].

The present study investigates useful Polish coal fly ash after chemical treatment with a 30% solution of $H_2O_2$ and assesses its abilities to remove toxic mixtures from the bi-component systems Cr(III)–CV and Cd(II)–CV. Multicomponent systems containing only metal ions or organic dyes have been studied previously. Industrial waste containing a mixture of components with different ionic radii poses a greater challenge. This manuscript will describe an important topic for future studies of the sorption and utilization of FA for the immobilization of Cr(III)–CV and Cd(II)–CV in bi-component systems from aqueous solutions.

## 2. Materials and Methods

### 2.1. Chemicals

The reagents used were of analytical reagent (AR) grade, produced by Avantor Performance Materials Poland S.A. (Gliwice, Poland), Merck KGaA (Darmstadt, Germany), and Sigma-Aldrich Corp. (St. Louis, MO, USA). Used reagents were as follows: 30% solution of $H_2O_2$, nitric(V) acid (a.p., POCh), sodium hydroxide, cadmium(II) nitrate(V), chromium(III) nitrate(V), crystal violet (CAS number: 548-62-9), standard cadmium and chromium solution (1000 $mg \cdot L^{-1}$). Cd(II), Cr(III) and CV stock solutions (1000 $mg \cdot L^{-1}$) were made by dissolving exact amounts of $Cd(NO_3)_2$, $Cr(NO_3)_3$ and crystal violet in distilled water, respectively. The test solutions were prepared by diluting stock solutions with distilled water. The obtained concentration of heavy metals in test solutions varied between 10 and 500 $mg \cdot L^{-1}$, and the concentration of dye varied between 1 and 400 $mg \cdot L^{-1}$.

### 2.2. Fly Ash Sample and Its Modification

The coal fly ash sample was collected from the heat and power plant in Rzeszów, the branch of PGE Energia Ciepła S.A. (Rzeszów, Poland). On the basis of a combined content of $SiO_2$, $Al_2O_3$ and

Fe$_2$O$_3$ of over 70%, the coal fly ash was defined as Class F [36] according to the American Society for Testing Materials (ASTM C618) classification.

The oxidized form of coal fly ash was converted in the following way: the fly ash was treated with a 30% solution of H$_2$O$_2$ at a solution-to-fly-ash ratio of 10:1 in weight. The activation was carried out in a closed round-bottomed flask reflux condenser and mixed for 12 h at 100 °C. The obtained FA–H$_2$O$_2$ was decanted, filtered and repeatedly washed with distilled water until the pH of the filtrate dropped to the pH of distilled water. Next, the FA–H$_2$O$_2$ was dried in an electric dryer at a temperature of 105 °C for 24 h [15].

### 2.3. Sorption of Analytes

In total, 0.5 g of FA–H$_2$O$_2$ was added to 100 mL glass flasks, which were filled with a 50 mL metal ion and dye solution of varying concentrations (10–500 mg·L$^{-1}$). Then, the mixtures were subjected to agitation at pH 9.0 for 1.5 h (see Section 3.4.1). The method was described according to the work in [37].

### 2.4. Determination of Analytes

Crystal violet (CV) dye, which is also known as methyl violet 10B, basic violet or 3(N-[4-[bis[4-dimethyl-amino]-phenyl]-methylene]-2,5-cyclohexadien-1-ylidine]-N-methyl-methanaminium chloride), was used. The absorbance at a wavelength of 595 nm for different concentrations of 1, 3, 5, 7, 9, 11, 13 and 15 mg·L$^{-1}$ CV was measured by the spectrophotometric method.

Calibrant solutions containing 0.0, 3.5, 5.0 and 7.0 mg L$^{-1}$ Cr and 0.0, 0.5, 1.0, 1.5 and 2.0 mg L$^{-1}$ Cd were prepared by diluting a stock solution of 1000 mg L$^{-1}$ Cr and Cd. The concentration of metals in solution after sorption was determined by the flame atomic absorption spectrometry (FAAS) method at 359.4 and 228.8 nm wavelengths for Cr and Cd, respectively.

## 3. Results and Discussion

### 3.1. Surface Morphology and Chemical Composition of FA and FA-H$_2$O$_2$

Scanning electron micrographs (SEM) show the surface morphology of raw FA (Figure 1a) and activated FA–H$_2$O$_2$ (Figure 1b). The SEM image of raw FA shows that the fly ash particles are generally spherical in shape (Figure 1a). According to Styszko-Grochowiak et al. [38], the FA occurs in the form of small, spherical grains, which consist mainly of mixed aluminosilicates and calcium. The pictograph of FA–H$_2$O$_2$ (Figure 1b) indicates that the modification process by H$_2$O$_2$ did not lead to crystallization and that the H$_2$O$_2$ was attached to the FA surface [39].

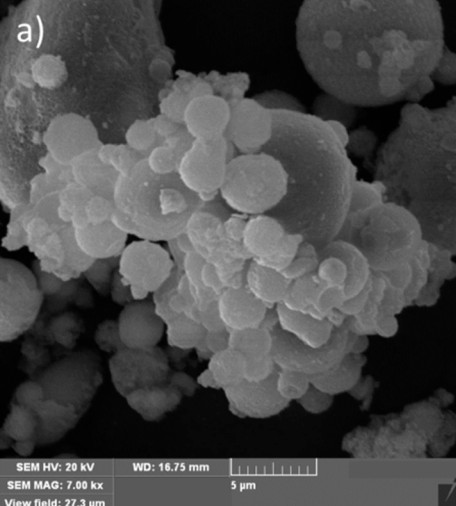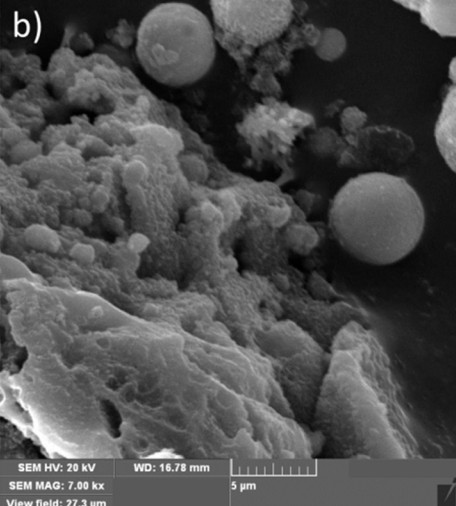

**Figure 1.** Scanning electron microscope (SEM) images of coal fly ash; (**a**) raw FA and (**b**) H$_2$O$_2$-treated FA.

The samples were detected by SEM equipped with energy-dispersive X-ray spectroscopy (EDS). For the detection of the elemental composition of FA and FA–$H_2O_2$, an analysis of X-ray signals was used (Figure 2). After a chemical treatment with hydrogen peroxide, the increase in oxygen content was shown to be owed to the oxidation of fly ash. In order to determine the percentages of carbon and oxygen, a spot analysis of a selected sample was carried out. The O/C ratio shows the effects of oxidation after the modification. As a result of FA activation with $H_2O_2$, the carbon at the surface is oxidized. The increase in the oxygen-to-carbon ratio in FA–$H_2O_2$ compared to raw FA supports this fact. The increase in the O/C ratio from 10.8 to 21.2 confirmed this (Figure 2).

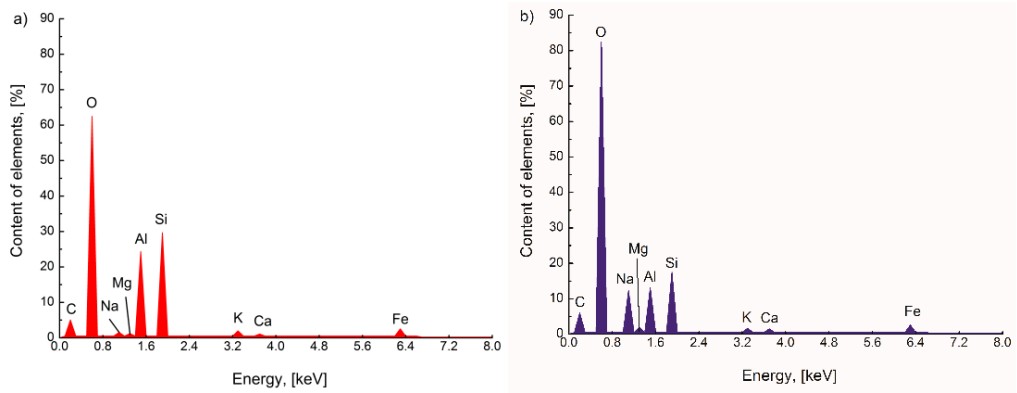

**Figure 2.** Energy dispersive spectroscopy (EDS) spectra of coal fly ash; (**a**) raw FA and (**b**) $H_2O_2$-treated FA.

The presence of quartz, mullite and magnetite [40] (Figure 3) is shown by the XRD patterns of FA and FA–$H_2O_2$. The diffraction pattern of FA–$H_2O_2$ indicated that the amounts of mineral in the FA–$H_2O_2$ increased in comparison to FA (higher than that of FA), as was observed in the height of their peak intensities.

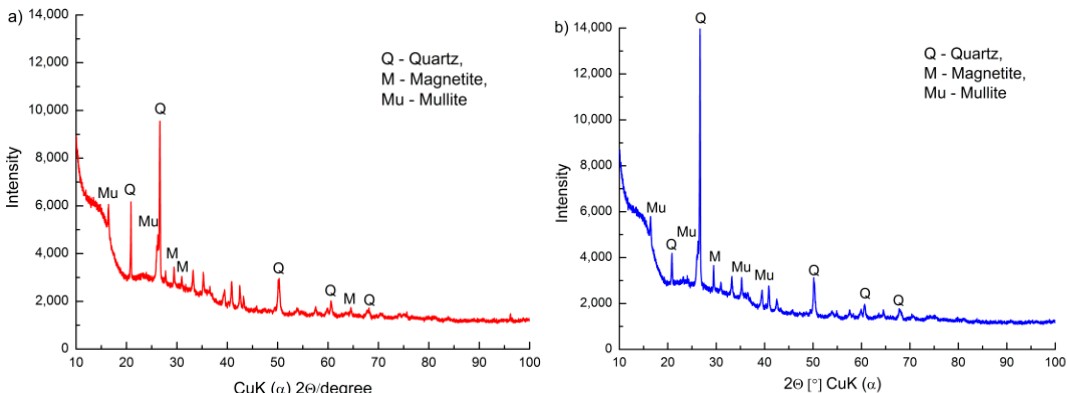

**Figure 3.** X-ray diffraction (XRD) patterns of (**a**) raw FA and (**b**) $H_2O_2$-treated FA.

### 3.2. FT-IR Measurements of FA and FA-$H_2O_2$

In the obtained Fourier transform infrared (FT-IR) spectra of FA and FA–$H_2O_2$, there was an observed absorption broadband at 3440 cm$^{-1}$, a band at 1630 cm$^{-1}$, and an absorption broadband at about 1050 cm$^{-1}$ (Figure 4, Table 1).

**Table 1.** The vibration bands of Fourier transform infrared (FT-IR) spectra [15,41–43].

| Adsorption Bands | Assignment | Interpretation |
|---|---|---|
| $3500–3000\ cm^{-1}$ | Stretching (–OH) and bending (H–O–H) vibrations | Water molecules adsorbed on the fly ash surface |
| $2920\ and\ 2850\ cm^{-1}$ | Stretching vibrations of –OH | Presence of hydrated aluminosilicates |
| $1630\ cm^{-1}$ | Bending vibrations of –OH and bending vibrations of H–O–H | |
| $1450\ cm^{-1}$ | Asymmetric stretching vibrations of bridge bounds C–O | Carboxyl–carbonate structures |
| $1050\ cm^{-1}$ | Asymmetric stretching vibrations of bridge bounds Si–O–Si and Si–O–Al | Tetrahedral or aluminum and silicon–oxygen bridges, typical for aluminosilicate framework structures |
| $800\ cm^{-1}$ | Symmetric stretching vibration Al, Si–O | Quartz |

Coal fly ash oxidized by $H_2O_2$ was more hydrophilic than that of the corresponding unoxidized sample [42,43]. There is a noticeable increase in band intensity in the field of $3440\ cm^{-1}$ in the spectrum of FA–$H_2O_2$.

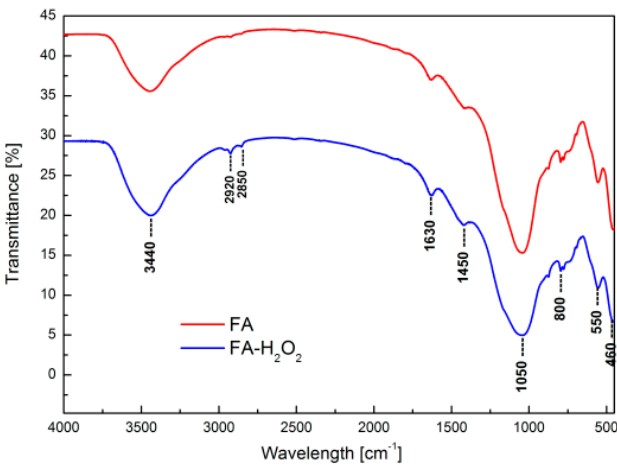

**Figure 4.** FT-IR spectra of raw FA and $H_2O_2$-treated FA.

The spectra of FA and FA–$H_2O_2$ contain a characteristic doublet of quartz at 798 and 780 $cm^{-1}$. Furthermore, the symmetrical stretching variation at 730 $cm^{-1}$ and the symmetrical stretching variation at 660 $cm^{-1}$ associated with Si–O–Al and Si–O–Si, respectively, were suggested by Mozgawa et al. [43]. Moreover, the band at about 550 $cm^{-1}$ corresponds to symmetric stretching vibrations of $\nu_s$ Si–O–Si bridge bonds and bending vibrations of $\delta$ O–Si–O as a complex band [15,42,43]. Based on the FT-IR spectrum, the presence of pore openings corresponding to the band at 460 $cm^{-1}$ in FA and FA–$H_2O_2$ can be observed, which can be attributed to the dissolution of the minerals (viz., quartz and mullite) present in the samples [15].

### 3.3. Thermal Gravimetric Analysis of FA and FA-$H_2O_2$

The curves of the differential thermal analysis (DTA) and thermogravimetric analysis (TG) of the FA sample, shown in Figure 5a, present a small mass loss in the temperature range from 50 to 150 °C, which is caused by the evaporation of water from the FA (Table 2).

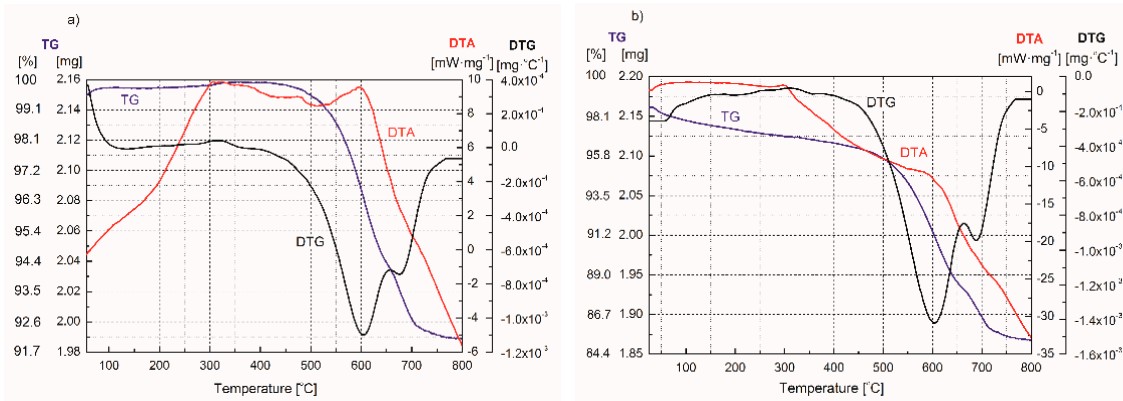

**Figure 5.** Differential thermal analysis (DTA), thermogravimetric analysis (TG) and differential thermogravimetric analysis (DTG) curves of (**a**) raw FA and (**b**) $H_2O_2$-treated FA.

This endothermic process for FA is assigned to the dehydration/thermodesorption of water, at around 0.2% [35]. The registered mass change in the temperature range from 250 to 400 °C indicates that the exothermic process is related to the oxidation of residue of organic matter, at about 7.4%. A larger mass loss in the temperature range of 400–600 °C is caused by the dehydration of calcium hydroxide (endothermic process) and the decomposition of calcium carbonate (exothermic process). The next dehydration process corresponds to the loss of coordinated and structural -OH groups of water (peak at 520 °C). This endothermic peak is assigned to the dehydroxylation of FA; i.e., the removal of hydroxyl groups from tetrahedral structural units. The next exothermic peak at the temperature of 600 °C indicates the recrystallization process. The last stage in the temperature range from 650 to 800 °C is the endothermic decomposition of the mineral structure. The total mass loss during the process amounted to 8%. The differential thermal analysis of FA–$H_2O_2$ (Figure 5b) indicates that the temperature range of 300–600 °C results in an endothermic process, which is caused by the decomposition of calcium hydroxide, calcium carbonate and oxygen functional groups on the surface of sample (Table 1).

**Table 2.** Changes in the properties of FA and FA–$H_2O_2$, which occurred as a result of heating during the thermogravimetric analysis.

| Process | Type of Reaction | Range of Reaction Temperature (°C) |
|---|---|---|
| Evaporation of water (moisture, hydration water) | Endothermic | 50–150 |
| Oxidation of residue of organic matter | Exothermic | 250–400 |
| Dehydration of calcium hydroxide | Endothermic | 400–600 |
| Decomposition of calcium carbonate | Exothermic | 400–600 |
| Dehydration of coordinated and structural water/dehydroxylation of FA and FA–$H_2O_2$ | Endothermic | 520 |
| Recrystallization | Exothermic | 600 |
| Decomposition of mineral structure | Endothermic | 650–800 |

## *3.4. Sorption Studies*

### 3.4.1. Influence of pH

The order of affinity results from different physical and electrostatic forces. These forces depend on the molecular size of sorbates. The sample activated by $H_2O_2$ has a greater oxygen surface than that of the corresponding raw coal fly ash. The presence of polar groups on the surface of FA–$H_2O_2$ is likely to give a considerable cation exchange capacity to the sorbents. The point of zero charge (PZC) was determined using the suspension and potentiometric titration methods [44,45]. In the case of FA/FA–$H_2O_2$, the value was found to be about 8.1/8.3. Therefore, the adsorbent surface has a negative charge at pH > 8.1/8.3 and positive charge at pH < 8.1/8.3. At a high pH, the solution in contact with

the FA–H$_2$O$_2$ contains excess hydroxyls. A combined process of precipitation (Cr(OH)$_3$, Cd(OH)$_2$) and adsorption (Cr(OH)$_4^-$) at pH > 9 was observed (Figure 6a,b). The order of affinity based on the amount of sorbate removal is as follows, with a hydrated ion radius: Cr(III) (4.13 Å) > Cd(II) (4.26 Å) > CV (above 40 Å) [37]. It was found that the sorption process predominates between positively charged CV molecules and the negatively charged silanol group. In the higher range of pH, the FA–H$_2$O$_2$ surface has a negative charge. Basically, crystal violet produced molecular cations (C$^+$) and reduced ions (CH$^+$) (Figure 6c). With the increasing solution pH, the sorption capacity increased due to the weaker forces of repulsion between the FA–H$_2$O$_2$ and CV.

The Cr(III) or Cd(II) (Cr(III)/Cd(II)) ions and CV dye sorption is attributed to different mechanisms, as shown in Figure 7.

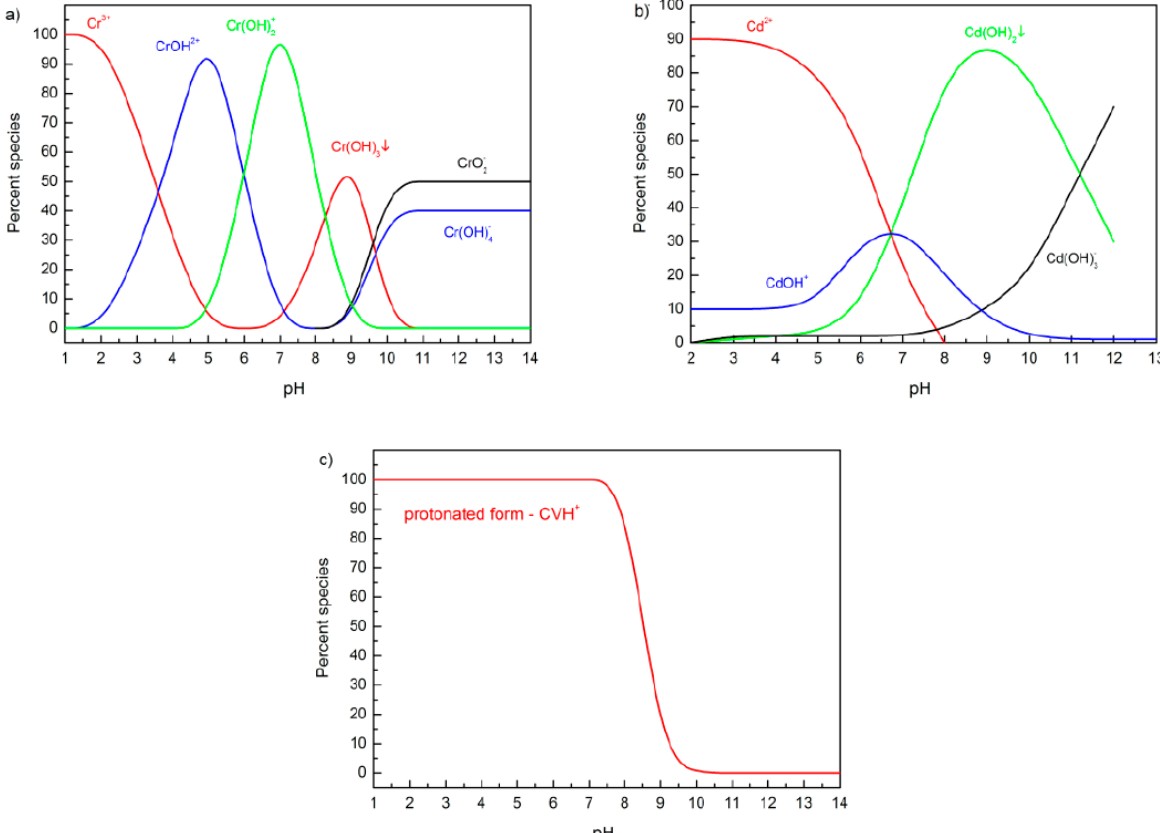

**Figure 6.** Distribution of the prevalent species of (**a**) chromium, (**b**) cadmium and (**c**) crystal violet in aqueous solution as a function of pH. The calculations were carried out using Visual Minteq ver. 3.0 (software of Jon Petter Gustafsson, KTH, Sweden).

Thus, a favorable pH for the sorption of cationic dyes, such as CV and Cr(III)/Cd(II) ions, will be greater than pH 9. The mechanism of Cr(III)/Cd(II) ions and CV dye sorption includes ion-exchange and surface adsorption processes and also dissolution–reprecipitation in the case of Cr(III) and Cd(II).

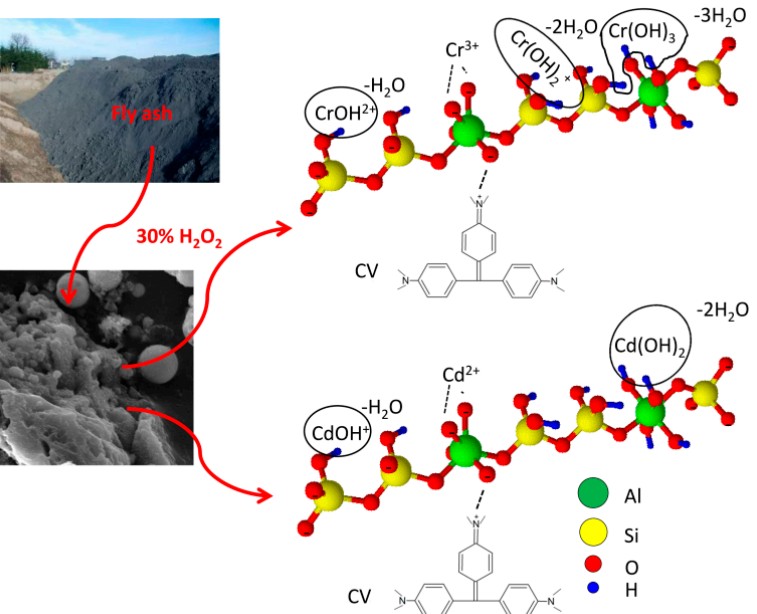

**Figure 7.** Schema of different mechanisms of the sorption process of Cr(III), Cd(II) and crystal violet (CV) onto FA–H$_2$O$_2$.

### 3.4.2. Effect of Initial Concentration

The values of adsorption (%) increased with the decrease in the initial Cd(II) ions and CV concentrations (Figure 8). The method was described according to that shown in [37].

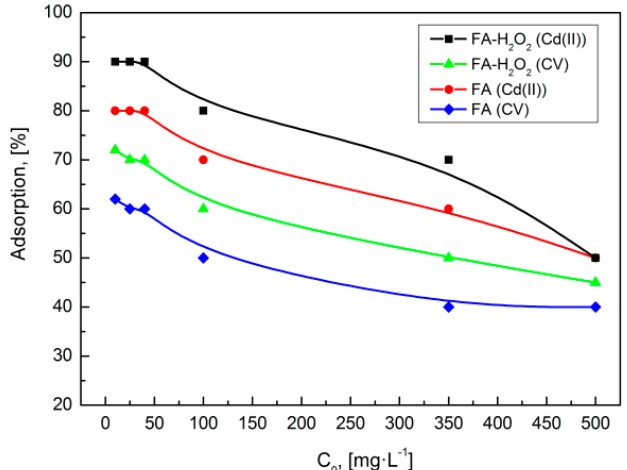

**Figure 8.** Effect of initial Cd(II) ion and CV concentration on the adsorption of FA and FA–H$_2$O$_2$.

### 3.4.3. Adsorption Isotherms in a Bi-Component System

The sorption isotherm models in a bi-component system were analyzed by means of the extended Langmuir (EL) [46], extended Langmuir–Freundlich (ELF) [47], and Jain–Snoeyink (JS) [48] models. The equations of the analyzed sorption isotherms are shown in Figure 9 and Table 3.

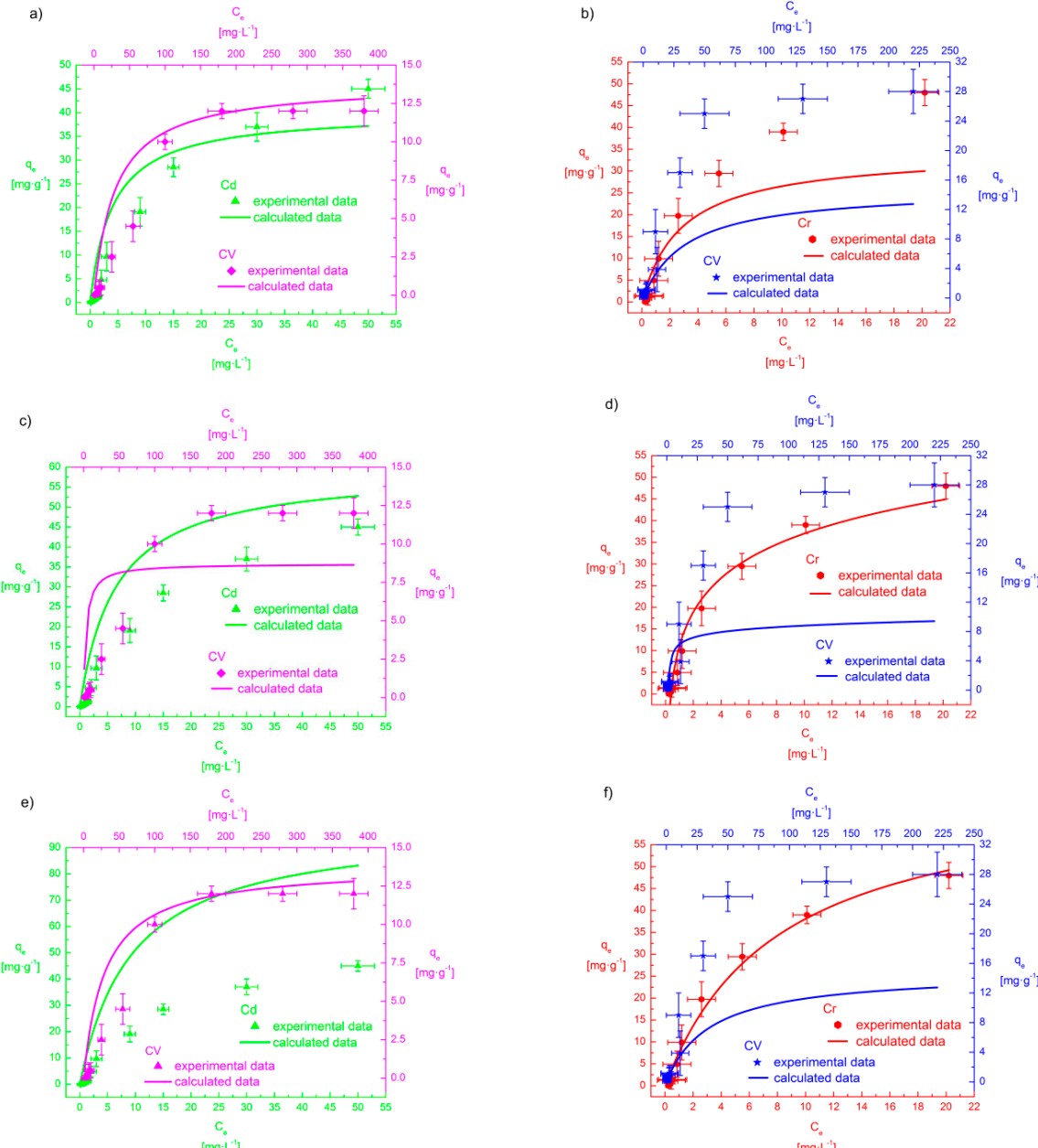

**Figure 9.** Estimation of Cr(III)–CV and Cd(II)–CV adsorption using (**a**) extended Langmuir model (EL) in the Cd(II)–CV system, (**b**) extended Langmuir model (EL) in the Cr(III)–CV system, (**c**) extended Langmuir–Freundlich (ELF) in the Cd(II)–CV system, (**d**) extended Langmuir–Freundlich (ELF) in the Cr(III)–CV system, (**e**) Jain–Snoeyink (JS) isotherm models in the Cd(II)–CV system, (**f**) Jain–Snoeyink (JS) isotherm models in the Cr(III)–CV system.

**Table 3.** Parameter values of the bi-component isotherms for Cr(III)–CV and Cd(II)–CV on FA-$H_2O_2$.

| Isotherm Model | Equation [1] | Parameter | Cd(II)–CV | | Cr(III)–CV | |
|---|---|---|---|---|---|---|
| | | | Cd(II) | CV | Cr(III) | CV |
| Extended Langmuir | $q_{ei} = q_{maxi} \dfrac{K_{Li}C_{ei}}{1+\sum\limits_{j=1}^{N} K_{Lj}C_{ej}}$ | $\chi^2$/DoF | 0.9 | 0.8 | 27.1 | 31.5 |
| | | $R^2$ | 0.984 | 0.992 | 0.773 | 0.698 |
| Extended Langmuir–Freundlich | $q_{ei} = q_{maxi} \dfrac{(K_{Li}C_{ei})^{1/n_i}}{1+\sum\limits_{j=1}^{N} \left(K_{Lj}C_{ei}\right)^{1/n_j}}$ | $\chi^2$/DoF | 33.3 | 45.8 | 47.9 | 0.6 |
| | | $R^2$ | 0.399 | 0.559 | 0.352 | 0.994 |
| Jain–Snoeyink | $q_{e1} = (q_{max1} - q_{max2})\dfrac{K_{L1}C_{e1}}{1+K_{L1}C_{e1}} +$ $+q_{max2}\dfrac{K_{L1}C_{e1}}{1+K_{L1}C_{e1}+K_{L2}C_{e2}}$ when $q_{max1} > q_{max2}$ $q_{e2} = q_{max2}\dfrac{K_{L2}C_{e2}}{1+K_{L1}C_{e1}+K_{L2}C_{e2}}$ | $\chi^2$/DoF | 36.7 | 0.04 | 0.2 | 34.6 |
| | | $R^2$ | 0.297 | 0.996 | 0.997 | 0.402 |

[1] where $q_{ei}$ is the uptake of component $i$ in the multicomponent system, $C_{ej}$ ($j = 1, 2, \ldots, N$; where $N$ is the number of components) is the equilibrium concentration of Cr(III), Cd(II) and CV in the system, $q_{maxi}$ and $K_{Li}$ are Langmuir constants, and the $n_i$ parameter is the heterogeneity factor. The parameters $q_{maxi}$, $K_{Li}$ and $n_i$ were obtained from the single isotherms for Cr(III), Cd(II) and CV and substituted in equals to predict the uptake of Cr(III), Cd(II) and CV in the multicomponent system. The maximum adsorption capacities obtained from the single isotherm ere found to be 775, 570 and 433 mg·g$^{-1}$ for Cr, Cd and CV, respectively.

In order to quantitatively compare the applications of each model, the coefficients of determination ($R^2$) and a reduced chi-square test ($\chi^2$/DoF) were calculated [49]. The linear and non-linear regression (reduced chi-square) value was calculated using the Origin Pro 7.5 software (OriginLab Corp., Northampton, MA, USA): $\chi^2/DoF = \frac{1}{DoF}\sum\limits_{i=1}^{N} \frac{(q_{ei}-q_{ei,m})^2}{q_{ei,m}}$, where $DoF$, $q_{ei}$ and $q_{ei,m}$ are degrees of freedom, experimental data and model data, respectively. The error bars in the line graph represent a confidence interval of t-distribution; the significance was set at $p = 95\%$ and $n = 3$.

Figure 10a,b shows the comparison of the experimental adsorption data of Cr(III)/CV and Cd(II)/CV onto FA–$H_2O_2$ and the calculated equilibrium sorption capacity $q_{ei}$ (mg·g$^{-1}$) values in the Cr(III)–CV and Cd(II)–CV systems, by means of the bi-component isotherm models mentioned previously. The dissimilarities in values of the Cr(III)–CV and Cd(II)–CV systems between the experimental and calculated values for the entire data set of Cr(III)/CV and Cd(II)/CV are shown in Table 3.

If most of the data points are placed around the 45° line, then the studied isotherm models could well represent the experimental adsorption data of the bi-component systems [46,50]. In the bi-component system Cd(II)–CV, the EL and JS models fitted well to the adsorption data of Cd(II)/CV and CV, respectively, onto FA–$H_2O_2$, and can be satisfactorily used to estimate equilibrium data from the single isotherm parameters. The JS model poorly fitted the equilibrium data in the Cd(II)–CV system for Cd(II) and could not be used to forecast the binary-system adsorption, because in this case there were many points lying far away from the 45° line, rather than being placed around this line [46]. The JS model fitted well to the equilibrium data in the Cr(III)–CV system for Cr(III). However, the JS bi-component model approximated well to the experimental data since the isotherm provided a $\chi^2$/DoF of 0.2, respectively, for Cr(III). The values of $\chi^2$/DoF indicated that the ELF bi-component isotherm fitted worst to the experimental data with the highest reduced chi-square tests of 33.3/45.8 and 47.9, respectively, for Cr(III)/CV and Cd(II), in comparison to the other investigated models. The ELF bi-component model approximated well to the experimental data since the isotherm provided a $\chi^2$/DoF of 0.6, respectively, for CV in the Cr(III)–CV system. The explanation for these results is that the isotherm does not incorporate any parameter accounting for the interactions between metal ion and cationic dye. The EL bi-component isotherm approximated reasonably well to the experimental data since the isotherm provided a $\chi^2$/DoF of 0.9 and 0.8, respectively, for Cd(II) and CV.

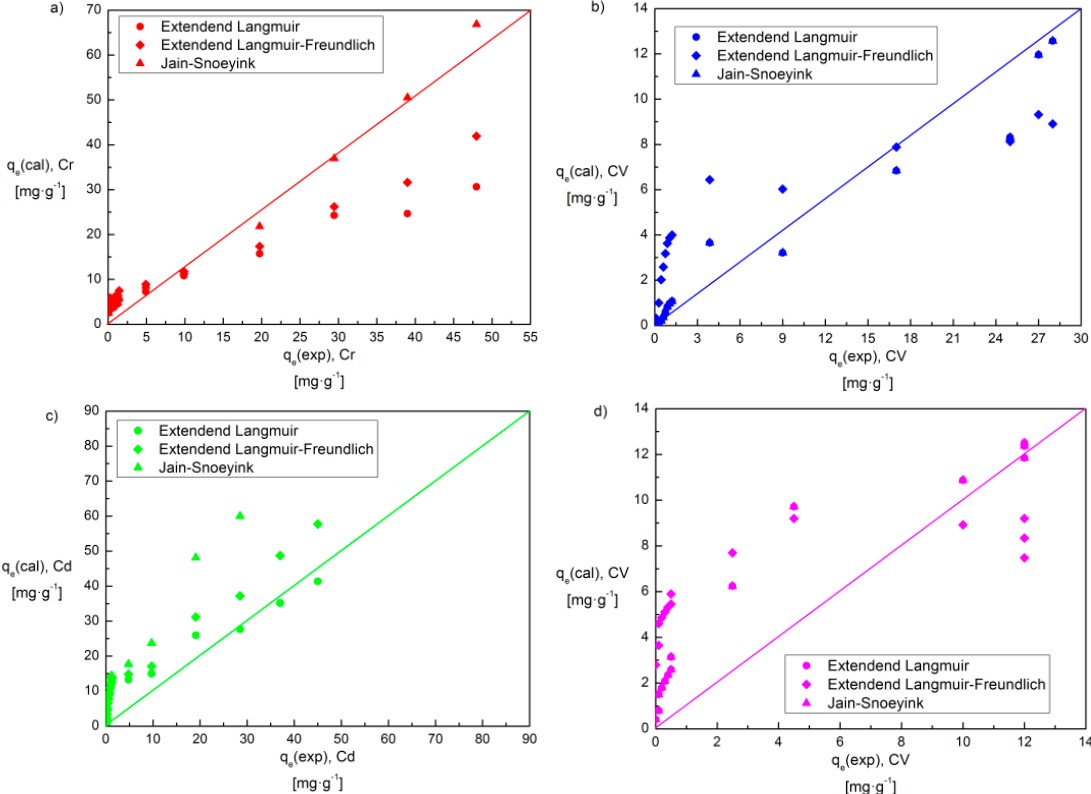

**Figure 10.** Scatter plot of estimated (cal.) vs. experimental (exp.) values for the equilibrium sorption capacity using the Extended Langmuir (EL), Extended Langmuir–Freundlich (ELF), Jain–Snoeyink (JS) isotherm models (**a**) qe (cal) Cr(III) vs. qe (exp) Cr(III) in the Cr(III)–CV system (**b**) qe (cal) CV vs. qe (exp) CV in the Cr(III)–CV system (**c**) qe (cal) Cd(II) vs. qe (exp) Cd(II) in the Cd(II)–CV system (**d**) qe (cal) CV vs. qe (exp) CV in the Cd(II)–CV system.

The results in the three-dimensional plot in Figure 11a–d show the metal or dye uptake (z-axis) as a function of the final equilibrium concentrations of these components (x- and/or y-axis, respectively). This method of presenting the experimental data was considered to represent parallel iso-concentration planes, which cut the area of the 3D plot [51]. The adsorption isotherm area of CV (Figure 11b,d) shows that the uptake of CV undergoes small changes concurrently with an increase of Cd(II)/Cr(III) at equilibrium. The adsorption isotherm area of Cd(II)/Cr(III) exhibits a very different shape (Figure 10a,b). The uptake of Cd(II)/Cr(III) was strongly decreased while the concentration of CV was increased. It is evident that, with the equilibrium concentration of CV changing from 0 to 200 mg·L$^{-1}$, the Cd(II)/Cr(III) adsorption capacity of FA–H$_2$O$_2$ was reduced by nearly 70%. The mesh areas determined by each sorbate and to the total metal or dye uptake, respectively, are according to the EL or JS model.

The adsorption experimental points in bi-component Cd(II)–CV and Cr(III)–CV systems for FA–H$_2$O$_2$ and the three-dimensional graphical surface plot [52,53] are shown as forecast by the EL model in Figure 11a,b, and the JS model in Figure 11c,d. Furthermore, the forecast of equilibrium data points with the isotherm using the EL and JS models was found to be satisfactory, respectively, for Cd(II)/CV and Cr(III)/CV. These models were calculated as the best fit for bi-component adsorption models based on the EL and JS isotherms for modeling the binary adsorption of Cd(II)/CV and Cr(III)/CV, respectively, from aqueous solutions for FA–H$_2$O$_2$ because it uses the single Langmuir isotherm parameters [52,53]. In this plot, the removal experimental points of Cd(II)/CV and Cr(III)/CV are shown together with the forecast isotherm using the EL and JS isotherms, respectively.

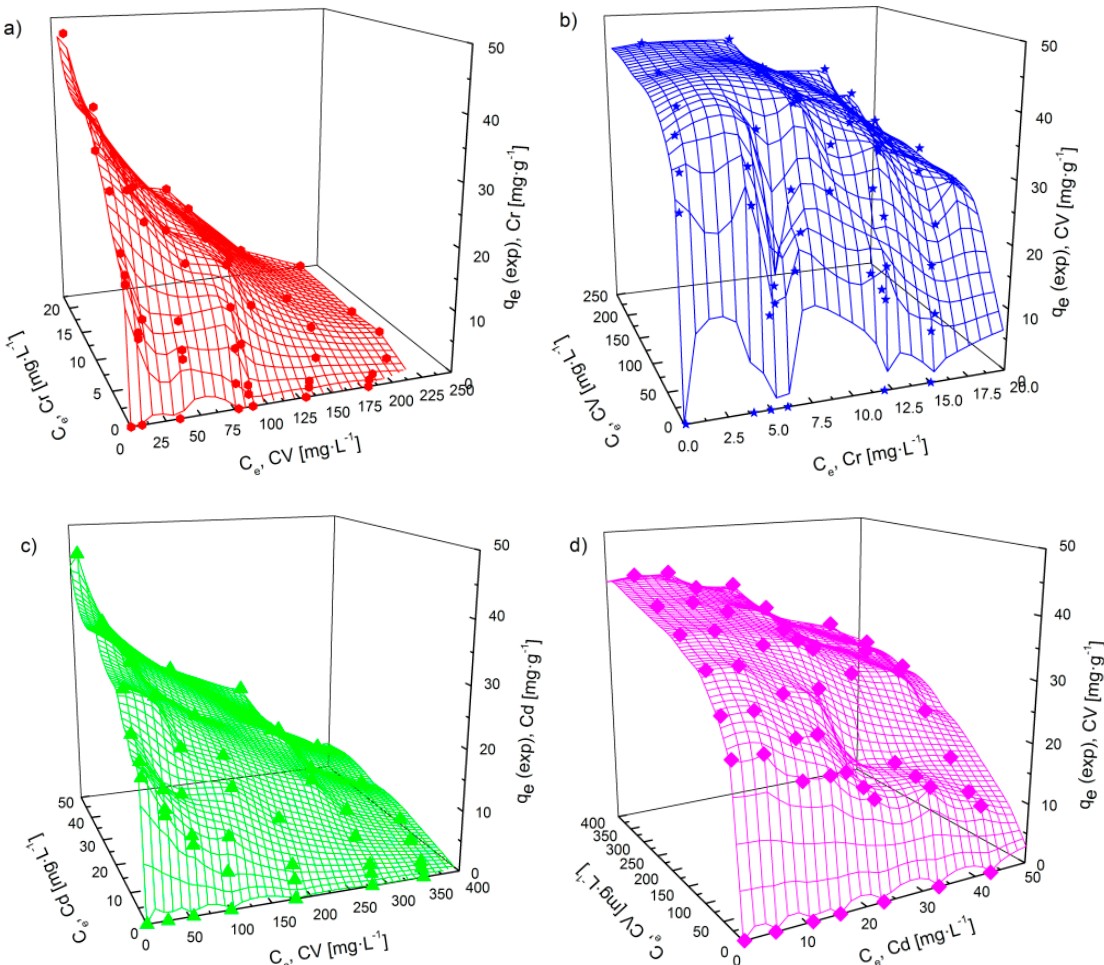

**Figure 11.** Forecast of surfaces by the EL model: (**a**) qe (exp) Cr(III) in the Cr(III)–CV system, (**b**) qe (exp) CV in the Cr(III)–CV system, (**c**) qe (exp) Cd(II) in the Cd(II)–CV system, and (**d**) qe (exp) CV in the Cd(II)–CV system.

Figure 11a shows the removal process of Cr(III) in the presence of CV, and in the range of CV concentration from 0 to 250 mg·L$^{-1}$. Figure 10c shows the removal process of Cd(II) in the presence of CV, and in the range of CV concentration from 0 to 400 mg·L$^{-1}$. The removal of Cr(III)/Cd(II) declined sharply while the concentration of CV increased at equilibrium. Figure 11b,c shows the dependence of Cr(III) and Cd(II) concentration on the removal of CV. The removal of CV was insignificantly constrained by the competition between Cr(III) and Cd(II), and the decrease in CV removal was nearly proportional to the increase in the Cr(III) and Cd(II) concentrations. In the bi-component systems, the removal of Cr(III) and Cd(II) was less than that of CV. Therefore, the CV showed a strong competitive effect in the adsorption of Cr(III) and Cd(II), while the Cr(III) and Cd(II) did not influence the adsorption of CV [52,53].

## 4. Conclusions

The results show that treating mineral by-products after the combustion of bituminous coal with H$_2$O$_2$ solution is a promising means to enhance the inhibition of Cr(III)–CV and Cd(II)–CV mobility in the bi-component system. In the first stage of the study, the characterization of FA and FA–H$_2$O$_2$ was carried out. The XRD data of FA and FA–H$_2$O$_2$ confirm the presence of minerals such as quartz, mullite and magnetite. The modification process by H$_2$O$_2$ did not cause crystallization, and the H$_2$O$_2$ was attached to the FA surface. The EL and JS models constitute the best interpretation of the bi-component system in the competitive adsorption of Cd(II)/CV and Cr(III)/CV, respectively. The competitive

adsorption in a single component system indicated that FA–$H_2O_2$ had a stronger effect on Cr(III) ions than on Cd(II) ions and CV dye, which is related to their hydrated ionic radius in the following order: Cr(III) > Cd(II) > CV. Based on the obtained adsorption capacity, the conclusion may be drawn that an $H_2O_2$-treated mineral by-product (FA–$H_2O_2$) can be used to treat pollutants containing Cr(III) ions and CV dye or Cd(II) ions and CV dye.

**Author Contributions:** Conceptualization, E.S.; methodology, E.S., D.P., and M.M.M.; validation, E.S., D.P., and M.M.M.; formal analysis, E.S., D.P., and M.M.M.; writing—original draft preparation, E.S.; writing—review and editing, E.S., D.P., and M.M.M.; visualization, E.S.; supervision, E.S., D.P., and M.M.M.; project administration, E.S., D.P., and M.M.M.; funding acquisition, D.P. All authors have read and agreed to the published version of the manuscript.

**Funding:** This research was funded by subsidies for statutory activity (number: DS.B0.19.001).

**Acknowledgments:** The X-ray analysis was carried out in the Laboratory of Spectrometry, Faculty of Chemistry, Rzeszów University of Technology, 6 Powstańców Warszawy Av., 35-959 Rzeszów, Poland and was financed from the DS budget.

**Conflicts of Interest:** The authors declare no conflict of interest.

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
