# Peer review of "Novel Application of Mineral By-Products Obtained from the Combustion of Bituminous Coal–Fly Ash in Chemical Engineering"

_minerals, doi:10.3390/min10010066_

Round 1

Reviewer 1 Report

Recently the discharge and disposal of heavy metal ions as well as carcinogen dyes have become urgent environmental issues. Among methods of their removal from industrial or mining effluents, adsorption has been gaining more and more attention due to its satisfactory efficiency and environmental friendliness, involving numerous typical adsorbents. Therefore in the paper the chemical modification of mineral by-products obtained from combustion of bituminous coal treated with hydrogen peroxide (30%) and its application in the removal of Cr(III) and Cd(II) ions and Crystal violet (CV) from the mixture of heavy metal and organic dye from one solution containing Cr(III)-CV or Cd(II)-CV were presented. Based on the FT-IR, TG, SEM-EDS and XRD analyses the mechanism of Cr(III)-CV or Cd(II)-CV sorption onto FA-H2O2 including ion-exchange and surface adsorption process was proposed. I think this paper is very interesting, the research results are very well documented. The references correspond to the content of the paper. However, in my opinion, the paper needs some changes (minor revision).

The introduction part is too long. Some paragraphs connected with the dyes, Cd, Cr should be shortened. As the calibration range for the heavy metal ions determination was presented, please add the calibration range for CV. Figs.2ab should be presented in the same scale. The same as Figs. 3ab The results presented in the 3D are very interesting, however, further analysis should be done. Please add some comments.

Author Response

12-th December, 2019.

Manuscript Minerals No: 661635

Title: Novel Application of Mineral by-products Obtained from Combustion of Bituminous Coal – Fly Ash in Chemical Engineering.

Authors: Eleonora Sočo 1,*, Dorota Papciak 2 and Magdalena Maria Michel 3

1  Department of Inorganic and Analytical Chemistry, Faculty of Chemistry, Rzeszów University of Technology, 6 Powstańców Warszawy Ave., PL-959 Rzeszów, Poland

2  Department of Water Purification and Protection, Faculty of Civil, Environmental Engineering and Architecture, Rzeszów University of Technology, 6 Powstańców Warszawy Ave., PL-959 Rzeszów, Poland

3  Institute of Environmental Engineering, Warsaw University of Life Sciences - SGGW, Nowoursynowska 166, Warsaw, 02-787, Poland

*Correspondence: e-mail: eleonora@prz.edu.pl, Tel.: +48-17-865-1508

In accordance with the Reviewer suggestion I respect the remarks.

Yours faithfully,

Eleonora Sočo

Dorota Papciak

Magdalena Maria Michel

Reviewer 2 Report

In this paper, the simultaneous sorption of Cr-Cv and Cr-Cd onto hydrogen peroxide treated fly ash. The synthetized material was characterized (SEM-EDX, XRD and FTIR). The subject treated is of undeniable importance because it deals with the valorization of a low cost and available industrial waste and the treatment of effluents containing chemicals whose toxicity is well known.

Experiments have been well conducted and the results obtained are of interest. Unfortunately, these results are sometime misinterpreted.

1- Review the language completely. I suggest that you have your manuscript read by a colleague using English as first language.

2- Include a chemicals sub-section in the Materials and Methods section, where the chemicals used can be described.

3- Page 3, line 98. In the experimental section, indicate the washing procedure applied to fly ash after reaction with hydrogen peroxide.

4- Page 3, line 114. Provide the signification of FAAS

Results and discussion

Several concerns in the material section

5- Some properties claimed by the authors such as mesoporosity (Surface Morphology and Chemical Composition of FA and FA-H2O2) cannot be obtained from SEM-EDX analysis. EDX is not accurate enough to be used as reliable quantitative tool.

6- The presence of bands at 2920 and 2850 cm-1 on the FTIR spectrum of FA-H2O2 should be explained

In the adsorption section

7- Figure 6 showing the speciation diagrams can be provided as supporting information, it would reduce the length of the manuscript.

8- Comparison of the performance of FA-H2O2 to that of FA would also improve the quality of the paper.

9- Generally, the fitting are too far from the experimental results, maybe the models were not selected properly?

Author Response

12-th December, 2019.

Manuscript Minerals No: 661635

Title: Novel Application of Mineral by-products Obtained from Combustion of Bituminous Coal – Fly Ash in Chemical Engineering

Authors: Eleonora Sočo 1,*, Dorota Papciak 2 and Magdalena Maria Michel 3

1  Department of Inorganic and Analytical Chemistry, Faculty of Chemistry, Rzeszów University of Technology, 6 Powstańców Warszawy Ave., PL-959 Rzeszów, Poland

2  Department of Water Purification and Protection, Faculty of Civil, Environmental Engineering and Architecture, Rzeszów University of Technology, 6 Powstańców Warszawy Ave., PL-959 Rzeszów, Poland

3  Institute of Environmental Engineering, Warsaw University of Life Sciences - SGGW, Nowoursynowska 166, Warsaw, 02-787, Poland

*Correspondence: e-mail: eleonora@prz.edu.pl, Tel.: +48-17-865-1508

In accordance with the Reviewer suggestion I respect the remarks.

Yours faithfully,

Eleonora Sočo

Dorota Papciak

Magdalena Maria Michel

Round 2

Reviewer 2 Report

Some of the comments were fully addressed.
Unfortunately, the quality of the language is still insufficient.
I thus recommend to accept the manuscript assuming that the language will be improve.

Author Response

Manuscript Minerals No: 661635

Title: Novel Application of Mineral By-Products Obtained from Combustion of Bituminous Coal – Fly Ash in Chemical Engineering

Authors: Eleonora Sočo 1,*, Dorota Papciak 2 and Magdalena Maria Michel 3

1  Department of Inorganic and Analytical Chemistry, Faculty of Chemistry, Rzeszów University of Technology, 6 Powstańców Warszawy Ave., PL-959 Rzeszów, Poland

2  Department of Water Purification and Protection, Faculty of Civil, Environmental Engineering and Architecture, Rzeszów University of Technology, 6 Powstańców Warszawy Ave., PL-959 Rzeszów, Poland

3  Institute of Environmental Engineering, Warsaw University of Life Sciences - SGGW, Nowoursynowska 166, Warsaw, 02-787, Poland

*Correspondence: e-mail: eleonora@prz.edu.pl, Tel.: +48-17-865-1508

In accordance with the Reviewer suggestion I respect the remarks.
